# What to expect of hardware metric predictors in NAS

**Kevin A. Laube**[1,2] **Maximus Mutschler**[1] **Andreas Zell**[1]

[1]Chair of Cognitive Systems, Eberhard-Karls-University Tuebingen, Germany
[2]Bosch Center for Artificial Intelligence

**Abstract** Modern Neural Architecture Search (NAS) focuses on finding the best performing architectures in hardware-aware settings; e.g., those with an optimal tradeoff of accuracy and latency. Due to many advantages of prediction models over live measurements, the search process is often guided by estimates of how well each considered network architecture performs on the desired metrics. Typical prediction models range from operation-wise lookup tables over gradient-boosted trees and neural networks, with little known information on how they compare. We evaluate 18 different performance predictors on ten combinations of metrics, devices, network types, and training tasks, and find that MLP models are the most promising. We then simulate and evaluate how the guidance of such prediction models affects the subsequent architecture selection. Due to inaccurate predictions, the selected architectures are generally suboptimal, which we quantify as an expected reduction in accuracy and hypervolume. We show that simply verifying the predictions of just the selected architectures can lead to substantially improved results. Under a time budget, we find it preferable to use a fast and inaccurate prediction model over accurate but slow live measurements. Code and results are available at `https://github.com/cogsys-tuebingen/NASLib`

## 1 Introduction

Modern neural network architectures are designed not only considering their primary objective, such as accuracy. While existing architectures can be scaled down to work with the limited available memory and computational power of, e.g., mobile phones, they are significantly outperformed by specifically designed architectures (Howard et al., 2017; Sandler et al., 2018; Zhang et al., 2018; Ma et al., 2018). Standard hardware metrics include memory usage, number of model parameters, Multiply-Accumulate operations, energy consumption, latency, and more; each of which may be limited by the hardware platform or network task. As the range of tasks and target platforms grows, specialized architectures and the methods to find them efficiently are gaining importance.

The automated design and discovery of specialized architectures is the main intent of Neural Architecture Search (NAS). This recent field of study repeatedly broke state of the art records (Zoph et al., 2018; Real et al., 2018; Cai et al., 2019; Tan and Le, 2019; Chu et al., 2019a; Hu et al., 2020) while aiming to reduce the researchers' involvement with this tedious and time-consuming process to a minimum. As the performance of each considered architecture needs to be evaluated, the hardware metrics need to be either measured live or guessed by a trained prediction model. While measuring live has the advantage of not suffering from inaccurate predictions, the corresponding hardware needs to be available during the search process. Measuring on-demand may also significantly slow down the search process and necessitates further measurements for each new architecture search. On the other hand, a prediction model abstracts the hardware from the search code and simplifies changes to the optimization targets, such as metrics or devices. The data set to train the predictor also has to be collected only once so that a trained predictor then works in the absence of the hardware it is predicting for, e.g., in a cloud environment. Furthermore, a differentiable predictor can be used for gradient-based architecture optimization of typically non-differentiable metrics (Cai et al., 2019; Xu et al., 2020; Nayman et al., 2021).

While the many advantages make predictors a popular choice of hardware-aware NAS (e.g. Xu et al. (2020); Wu et al. (2019); Wan et al. (2020); Dai et al. (2020); Nayman et al. (2021)), there are no guidelines on which predictors perform best, how many training samples are required, or what happens when a predictor is inaccurate. This work investigates the above points. As a first contribution, we conduct large-scale experiments on ten hardware-metric datasets chosen from HW-NAS-Bench (Li et al., 2021a) and TransNAS-Bench-101 (Duan et al., 2021). We explore how powerful the different predictors are when using different amounts of training data and whether these results generalize across different network architecture types. As a second contribution, we extensively simulate the subsequent architecture selection to investigate the impact of inaccurate predictors. Our results demonstrate the effectiveness of network-based prediction models; provide insights into predictor mistakes and what to expect from them. To facilitate reproducibility and further research, our experimental results and code are made available in the supplementary material.

## 2 Related work

**NAS Benchmarks**. As the search spaces of NAS methods often differ from one another and lack extensive studies, the difficulty of fair comparisons and reproducibility have become a major concern (Yang et al., 2019; Li and Talwalkar, 2020). To alleviate this problem, researchers have exhaustively evaluated search spaces of several thousand architectures to create benchmarks (Ying et al., 2019; Dong and Yang, 2020; Dong et al., 2020; Siems et al., 2020), containing detailed statistics for each architecture. TransNAS-Bench-101 (Duan et al., 2021) evaluates several thousand architectures across seven diverse tasks and finds that the best task-specific architectures may vary significantly.

The popular NAS-BENCH 201 benchmark (Dong and Yang, 2020) has been further extended with ten different hardware metrics for all 15625 architectures. Major findings of this HW-NAS Bench (Li et al., 2021a) include that FLOPs and the number of parameters are a poor approximation for other metrics such as latency, and that hardware-specific costs do not correlate well across hardware platforms. While accounting for each device's characteristics improves the NAS results, it is also expensive. Predictors can reduce costs by requiring fewer measurements and shorter query times. [1].

**Predictors in NAS**. Aside from real-time measurements (Tan et al., 2019; Yang et al., 2018), hardware metric estimation in NAS is commonly performed via Lookup Table (Wu et al., 2019), Analytical Estimation or a Prediction Model (Dai et al., 2020; Xu et al., 2020). While an operation- and layer-wise Lookup Table can accurately estimate hardware-agnostic metrics, such as FLOPs or the number of parameters (Cai et al., 2019; Guo et al., 2020; Chu et al., 2019a), they may be suboptimal for other metrics with non-obvious and non-linear factors that depend on hardware specifics. Such details can be captured with neural networks (Dai et al., 2020; Mendoza and Wang, 2020; Ponomarev et al., 2020; Xu et al., 2020) or other specialized models (Yao et al., 2018; Wess et al., 2021).

Of particular interest is the correct prediction of the model loss or accuracy, possibly reducing the architecture search time by orders of magnitude (Mellor et al., 2020; Wang et al., 2021; Li et al., 2021b). In addition to common predictors such as Linear Regression, Random Forests (Liaw et al., 2002) or Gaussian Processes (Rasmussen, 2003); specialized techniques may exploit training curve extrapolation, network weight sharing or gradient information. Our experiments follow the recent large-scale study of White et al. (2021), who compare 31 diverse accuracy prediction methods based on initialization and query time, using three NAS benchmarks.

## 3 Predicting hardware metrics

Our methods follow the large-scale study of White et al. (2021), who compared a total of 31 accuracy prediction methods. The differences between accuracy and hardware-metric prediction,

---

[1]For further reading, we recommend a recent survey on hardware-aware NAS (Benmeziane et al., 2021)

our selection of predictors, and the general training pipeline are described in this section. We then compare these predictors across different training set sizes in our experiments on HW-NAS-Bench and TransNAS-Bench-101, described in Section 4.

**Differences to accuracy predictors**. There are fundamental differences when predicting hardware metrics and the accuracy of network topologies. The most essential is the cost to obtain a helpful predictor, which may vary widely for accuracy prediction methods. While determining the test accuracy requires the costly and lengthy training of networks, measuring hardware metrics does not necessitate any network training. Consequentially, specialized accuracy-estimation methods that rely on trained networks, loss history, early stopping, gradient statistics or others do not apply to hardware metrics. Therefore, the dominant hardware-metric predictor family is model-based.

Since all relevant predictors are model-based, they can be compared by their training set size. This simplifies the initialization time of a predictor as the number of prior measured architectures on which they are trained. Since an untrained network and a few batches suffice to measure hardware-metrics, the collection of such a training set is comparably inexpensive.

Additionally, hardware predictors are generally used supplementary to a one-shot network optimized for loss or accuracy. Depending on the NAS method, a fully differentiable predictor is required in order to guide the gradient-based architecture selection. Typical choices are Lookup Tables (Cai et al., 2019; Nayman et al., 2021) and neural networks (Xu et al., 2020).

**Model-based predictors**. The goal of a predictor $f_p(a)$ is to accurately approximate the function $f(a)$, which may be, e.g., the latency of an architecture $a$ from the search space $\mathcal{A}$. A model-based predictor is trained via supervised learning on a set $\mathcal{D}_{train}$ of datapoints $(a, f(a))$, after which it can be inexpensively queried for estimates on further architectures. The collection of the dataset and the duration of the training are referred to as *initialization time* and *training time* respectively.

The quality of such a trained predictor is generally determined by the (ranking) correlation between measurements $\{f(a)|a \in \mathcal{A}_{test}\}$ and predictions $\{f_p(a)|a \in \mathcal{A}_{test}\}$ on the unseen architectures $\mathcal{A}_{test} \subset \mathcal{A}$. Common correlation metric choices are Pearson (PCC), Spearman (SCC) and Kendall's Tau (KT) (Chu et al., 2019b; Yu et al., 2020; Siems et al., 2020).

Our experiments include 18 model-based predictors from different families: Linear Regression, Ridge Regression (Saunders et al., 1998), Bayesian Linear Regression (Bishop, 2007), Support Vector Machines (Cortes and Vapnik, 1995), Gaussian Process (Rasmussen, 2003), Sparse Gaussian Process (Candela and Rasmussen, 2005), Random Forests (Liaw et al., 2002), XGBoost (Chen and Guestrin, 2016), NGBoost (Duan et al., 2020), LGBoost (Ke et al., 2017), BOHAMIANN (Springenberg et al., 2016), BANANAS (White et al., 2019), BONAS (Shi et al., 2020), GCN (Wen et al., 2020), small and large Multi-Layer-Perceptrons (MLP), NAO (Luo et al., 2018), and a layer-operation-wise Lookup Table model. Further details are provided in the supplementary material.

**Study setup**. We use the NASLib library (Ruchte et al., 2020) for a reliable and large-scale comparison. Each predictor was fit on each dataset and training size 50 times, using seeds $\{0, ..., 49\}$ and normalized input values. We also include a brief hyper-parameter optimization for every predictor. Details are provided in the supplementary material.

## 4 Predictor Experiments

We compare the different predictor models based on two NAS benchmarks, HW-NAS-Bench (Li et al., 2021a) and TransNAS-Bench-101 (Duan et al., 2021). They differ considerably by their network tasks, hardware devices, and architecture designs.

**HW-NAS-Bench architecture design and datasets**. In HW-NAS-Bench, each architecture is solely defined by the topology of a building block ("cell"), which is stacked multiple times to create a complete network. Each cell is completely defined by choosing six candidate operations. Since they

select from five different candidates each time, there are $5^6 = 15625$ unique cell topologies. These cells are not fully sequential but contain paths of different lengths.

HW-NAS-Bench provides ten hardware statistics on CIFAR10, CIFAR100 Krizhevsky et al. (2009) and ImageNet16-120 Chrabaszcz et al. (2017), of which we exclude the incomplete EdgeTPU metric. Thus there are 27 data sets of varying difficulty. As shown in the supplementary material, 12 of them can be accurately fit with Linear Regression and only 25 training samples. Many are also very similar since their measured networks differ only by the number of image classes. We therefore select five datasets that (1.) are not trivial to learn as they are non-linear and (2.) not redundant:

- ImageNet16-120, raspi4, latency
- CIFAR100, pixel3, latency
- CIFAR10, edgegpu, latency
- CIFAR100, edgegpu, energy consumption
- ImageNet16-120, eyeriss, arithmetic intensity

**TransNAS-Bench-101 architecture design and datasets**. TransNAS-Bench-101 contains information for 7,352 different network architectures, used as backbones in seven diverse vision tasks. Since 4,096 are also a subset of HW-NAS-Bench, we focus on the remaining 3,256 architectures with a macro-level search space. Unlike a micro-level search space, where a cell is stacked multiple times to create a network, each network layer and block is considered individually. In particular, the TransNAS-Bench-101 networks consist of four to six pairs of ResNet blocks (He et al., 2016), which may modify the image size and channels in four ways: not at all, double the channel count, halve the spatial size, and both. Every network has to double the channel count 1 to 3 times, resulting in 3,256 unique architectures. The networks may consequentially differ in their number of layers (depth), the number of channels (width), and image size at any layer.

As done for HW-NAS-Bench, we select five of the seven available datasets for their latency measurements.

- Object classification
- Scene classification
- Room layout
- Jigsaw
- Semantic segmentation

**Fitting results and comparison**. The results, averaged over all selected HW-NAS-Bench and TransNAS-Bench-101 datasets, are presented in Figures 1a and 1b, respectively. The left plots present the absolute predictor performance, the right ones make relative comparisons easier.

Unsurprisingly, more training samples (i.e., evaluated architectures) generally lead to better prediction results, even until the entire search space is known (aside from the test set). This is true for most of the predictors, although e.g. Gaussian Processes and BOHAMIANN saturate early. The simple Linear Regression and Ridge Regression models also fail to make proper use of hundreds of data points but perform decently when only a few training samples are available. Interestingly, the same is true for the graph-encoding network-based predictors BONAS and GCN. While knowing how the different paths within each cell connect is especially useful given less than fifty training samples, the advantage disappears afterward. In contrast, the graph-encoding encoder-decoder approach of NAO performs decently at all times.

Due to their powerful rule-based approach, tree-based models perform much better given many training samples. Under such circumstances, LGBoost is a candidate for the best predictor model. Similarly, the predictions of Support Vector Machines also benefit strongly from more samples.

The model we find to perform best for most training set sizes are MLPs. They are among the top predictors at almost all times in the HW-NAS-Bench, although tree-based models are competitive given enough data. After around 3,000 training samples, thinner and deeper MLPs improve over the wider and smaller ones. The path-encoding BANANAS model behaves similarly to a regular large MLP but requires more samples to reach the same performance. This is interesting since, aside from the data encoding, BANANAS is an ensemble of three large MLP models. Even though only the first network layer is affected by the data encoding, the more complicated path-encoding proves harmful when the connectivity of the architectures in the search space is fixed. On TransNAS-Bench-101,

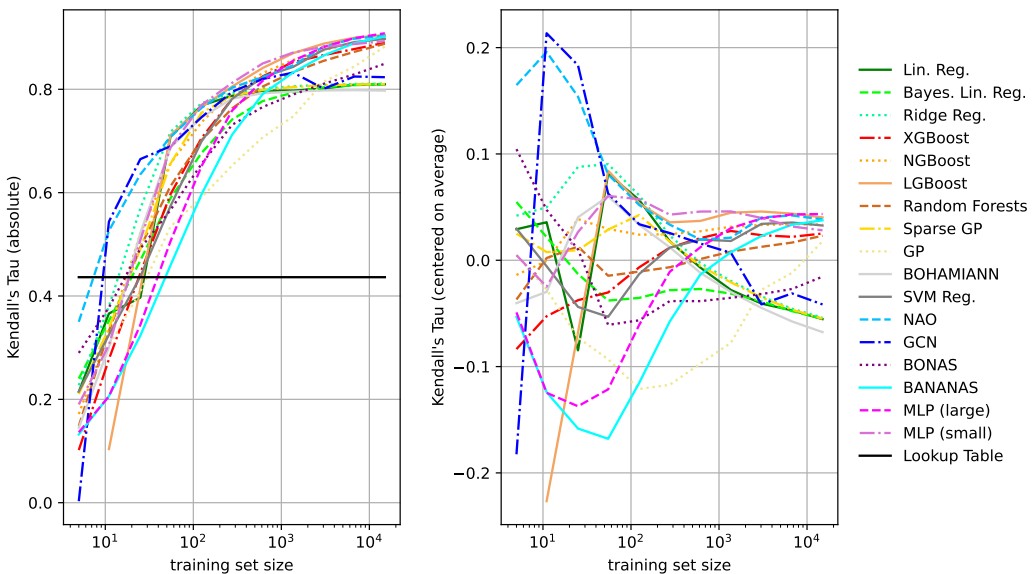

(a) Results on HW-NAS-Bench. NAO performs decently at all times, and none of the prediction models requires more than 60 training samples to improve over a Lookup Table model.

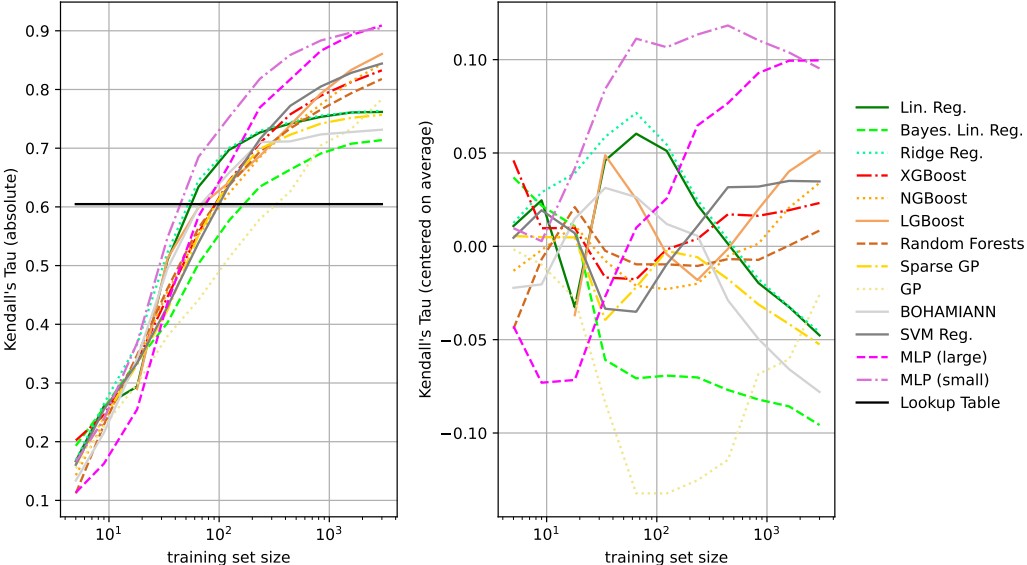

(b) Results on TransNAS-Bench-101. Since all network architectures are purely sequential by design, we do not evaluate predictors that specifically encode the architecture connectivity (BANANAS, BONAS, GCN, NAO). After as few as 20 training samples, MLP models outclass all other predictors.

Figure 1: How well the different predictors rank the test architectures, depending on the training set size and averaged over the five selected datasets. **Left plots**: absolute Kendall's Tau ranking correlation, higher is better. **Right plots**: same as left, but centered on the predictor-average.

MLP perform exceptionally well. They are much better than any other tested predictor once more than just 20 training samples are available. The small MLP model can achieve a KT correlation of 80% with just 200 training samples, which takes the best non-network-based predictor (Support Vector Machine) four times as many. They are also the only models that achieve a KT correlation of over 90%, about 5% higher than the next best model (LGBoost).

Finally, the Lookup Table models (black horizontal lines) perform poorly in comparison to any other predictor. Even though building such a model for HW-NAS-Bench datasets requires only 25 neighbored architectures, NAO and GCN perform better after just ten random samples. More than

Table 1: The Kendall's Tau correlation of Lookup Tables and Linear Regression (in brackets, using only 124 training samples) across metrics and devices.

| | HW-NAS-Bench | | | | | TransNAS-101 |
|---|---|---|---|---|---|---|
| | Raspi4 | FPGA | Eyeriss | Pixel3 | EdgeGPU | Tesla V100 |
| latency | 0.45 (0.75) | 0.99 (0.97) | 0.99 (0.96) | 0.49 (0.78) | 0.21 (0.79) | 0.60 (0.70) |
| energy | | 0.99 (0.97) | 1.00 (0.99) | | 0.23 (0.79) | |
| arithmetic_intensity | | | 0.84 (0.81) | | | |

half of the predictor models require less than 25 random samples, while the worst need at most 60. On TransNAS-Bench-101, Lookup Tables perform comparably better. Building one requires only 21 neighbored architectures, and it takes most models between 50 and 100 random training samples to achieve better performance. When measured on a per dataset basis, we find that the Lookup Table models display a severe performance difference ranging from about 20% KT correlation (cifar10-edgegpu_latency and Jigsaw) to over 70% (ImageNet16-120-eyeriss_arithmetic_intensity and Semantic Segmentation). Other models prove to be much more stable.

**Devices and Metrics**. The previously described results are based on a specific selection of HW-NAS-Bench and TransNAS-Bench-101 datasets that are hard to fit for Lookup Table models. As shown in Table 1, that is not always the case. The FPGA and Eyeriss hardware devices are very suitable for Lookup Tables, so that achieving an almost perfect ranking correlation is possible. Nonetheless, Linear Regression requires only 124 training samples to compete even there and is significantly better in every other case. We finally observe that the difficulty of fitting predictors primarily depends on the hardware device, much more than the measured metric.

## 5 Evaluating the predictor-guided architecture selection

Although the experiments in Section 4 greatly assist us in selecting a predictor, it is not clear what a specific Kendall's Tau correlation implies for the subsequent architecture selection. Given a perfect hardware metric predictor (Kendall's Tau = 1.0), we can expect that an ideal architecture search process will select the architectures with the best tradeoff of accuracy and the hardware metric, i.e., the true Pareto front. On the other hand, imperfect predictions result in the selection of supposedly-best architectures that are wrongly believed to be better.

To study how hardware predictors affect NAS results, we extensively evaluate the selection of such supposedly-best architectures in simulation. This approach can evaluate any combination of predictor quality, test set size, and dataset, without the technical difficulties of obtaining actual predictor models that precisely match such requirements. Since the hardware and accuracy prediction models are usually independent and can be studied in isolation, we use ground-truth accuracy values in all cases.

**Simulating predictors**. The main challenge of the simulation is to quickly and accurately model predictor outputs. We base our simulation on how predictor-generated values deviate from their ground-truth targets on the test set, which is explained in Figure 2 and further detailed in the supplementary material. Since the simulated deviations are similar to those of actual predictors, simulated predictions are obtained by drawing random values from this deviation distribution and adding them to the ground-truth hardware measurements.

A single example of a simulation can be seen in Figure 3. Although most selected architectures (orange) are close to the true optimum (red Pareto front), there almost always exists an architecture that has superior accuracy and, at most, the same latency. Simply accepting the 13 selected architectures in this particular example results in a mean reduction of accuracy ($MRA_{all}$) of 1.06%. In other words, the average selected architecture has 1.06% lower accuracy than a comparable one

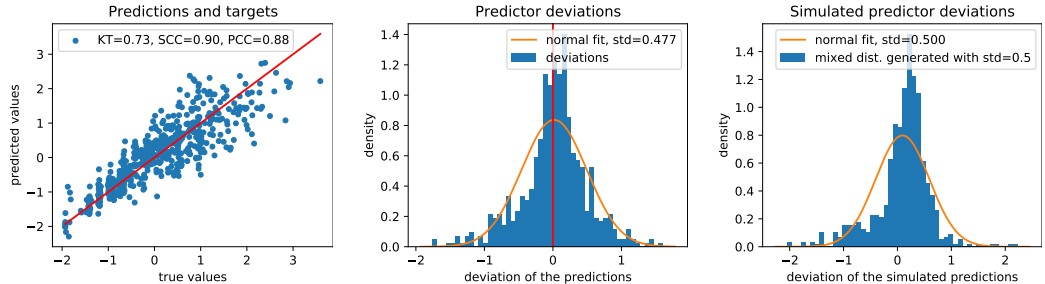

Figure 2: A trained XGBoost prediction model on normalized ImageNet16-120 raspi4-latency test data. **Left**: The latency prediction (y-axis) for every architecture (blue dot) is approximately correct (red line). **Center**: The same data as on the left, the distribution of deviations made by the predictor (blue) and a normal distribution fit to them (orange). **Right**: A mixed distribution can simulate typical deviation distributions as that in the center plot.

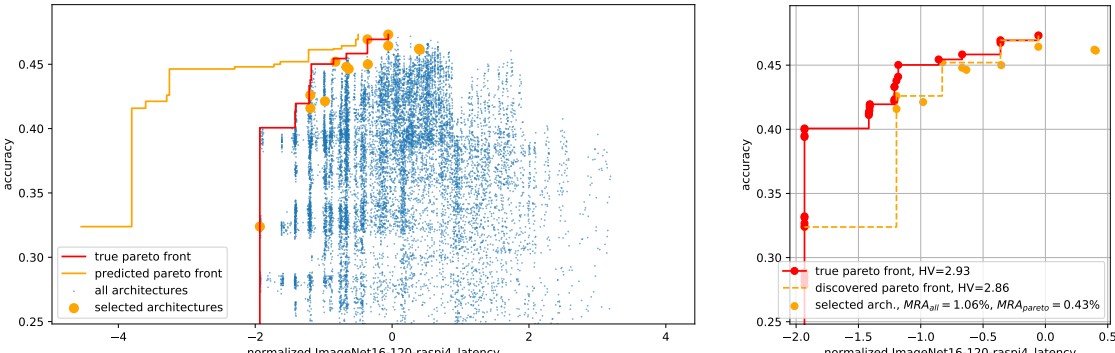

Figure 3: An example of predictor-guided architecture selection, std=0.5. **Left**: The simulated predictor makes an inaccurate latency prediction for each architecture (blue), resulting in the selection of the supposedly-best architectures (orange dots). Even though the predicted Pareto front (orange line) may differ significantly from the ground-truth Pareto front (red line), most selected architectures are close to optimal. **Right**: Same data as left. Simply accepting all selected architectures results in a Mean Reduction of Accuracy (*MRA*) of 1.06%, while verifying the predictions and discarding inferior results improves that to 0.43%. The hypervolume (HV, area under the Pareto-fronts) is reduced by 0.07.

on the true Paret front. However, simply verifying the hardware metric predictions through actual measurements reveals that some selected architectures are suboptimal. By choosing only the Pareto subset of the selection, the opportunity loss can be reduced to 0.43% ($MRA_{pareto}$).

An important property of this approach is that it is independent of any particular optimization method. The supposedly-best architectures are always correctly identified, which is an upper bound on how well Bayesian Optimization, Evolutionary Algorithms, and other approaches can perform. The exemplary $MRA_{pareto}$ opportunity loss of 0.43% is therefore *unavoidable* and depends solely on the hardware metric predictor, the dataset, and the number of considered architectures.

**Results.** We simulate 1,000 architecture selections for each of the five chosen HW-NAS-Bench datasets, six different test set sizes, and eleven distribution standard deviations between 0.0 and 1.0. As exemplarily shown in Figure 3, each such simulation allows us to compute the mean reduction in accuracy (MRA) and the hypervolume (HV) under the Pareto fronts. The most important insights are visualized in Figure 4 and summarized below.

Verifying the predicted results matters (Figure 4, left). The best prediction models achieve a KT correlation of almost 0.9, which translates to a mean reduction in accuracy of $MRA_{all} \approx 1.5\%$. That means, for each selected architecture, there exists an architecture of equal or lower latency in the true Pareto set (if latency is the hardware metric) that improves the average accuracy by 1.5%.

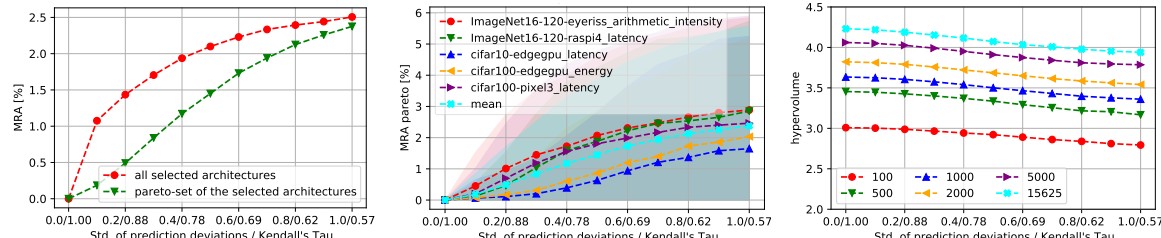

Figure 4: Simulation results, with the standard deviation of the predictor deviations and the resulting KT correlation on the x-axis. **Left**: Verifying the hardware predictions can significantly improve the results, even more so for better predictors. **Center**: The drops in average accuracy are dependant on the dataset and hardware metric. **Right**: Considering more candidate architectures and using better prediction models results in a larger hypervolume.

Even though all selected architectures are believed to form a Pareto set, that is not the case. Their optimal subset has a reduction of only $MRA_{pareto} \approx 0.5\%$, a significant improvement. However, finding this optimal subset requires actually measuring the hardware metrics of the architectures selected by the used NAS method.

Furthermore, the left of Figure 4 aids in anticipating the MRA given a specific predictor. If one used e.g. BOHAMIANN (KT≈0.8, see Figure1a) instead of MLPs or LGBoost (KT≈0.9), $MRA_{pareto}$ increases from around 0.5% to roughly 1.2%. The average accuracy of the selected architectures is thus reduced by another 0.7%, just by using an unsuitable hardware metric predictor. Lookup Tables (KT≈0.45) are not even visualized anymore, they have an $MRA_{pareto}$ of over 2.5%.

Another interesting observation is that the gap between $MRA_{all}$ and $MRA_{pareto}$ is wider for better predictors. This is a shortcoming of the MRA metric that we elaborate on in the supplementary material.

The dataset and metric matter (Figure 4, center). While we generally present the results averaged over datasets, there exists some discrepancy among them. Most interestingly, predicting hardware metrics on harder classification problems (ImageNet16-120 is harder than CIFAR10) also results in a higher MRA. This is especially important since MRA is an absolute accuracy reduction. Even though the CIFAR10 networks achieve twice the accuracy of ImageNet16-120 networks, they lose less absolute accuracy to imperfect predictions. The order of MRA/dataset is primarily stable for any predictor KT correlation. Finally, as visualized by the shaded areas, the standard deviation of the MRA is huge. Consequentially, predictor-guided NAS is very likely to produce results of varying quality for each different predictor or search attempt, especially with less accurate predictors.

The number of considered architectures matters (Figure 4, right). We measure the hypervolume of the discovered Pareto front (i.e., the area beneath it), which, unlike MRA, also considers the hardware metric. Quite obviously, if the architectures from the true Pareto set are not considered, they can not be selected. To achieve the highest possible hypervolume of around 4.2 (i.e. find the true Pareto set), every architecture in the search space must be evaluated with a perfect predictor. This is impossible in most real-world cases, where only a tiny fraction of all possible architectures can ever be considered.

For HW-NAS-Bench, considering 5000 architectures with perfect live measurements and predicting the metrics for all 15625 with ranking correlation KT≈0.73 results in choosing architecture sets of similar performance, measured by their hyper-volume. As seen in Figure 1a, Ridge Regression can achieve this performance with fewer than 100 training samples. Thus, a worse predictor leads to better results if it enables considering more architectures. This insight is especially crucial for live measurements, which are accurate but slow. Similarly, estimating the network accuracy with super-networks takes much more time than predicting their performance with a neural pre-

dictor (Wen et al., 2020). If the measurement of any metrics is the limiting factor, a guided selection of a cheap predictor is likely to do better.

# 6 Discussion

**Chosen prediction methods**. Given the nature of hardware-metric prediction, only the subset of model-based predictors evaluated by White et al. (2021) is suitable. We extended this subset with four models, including the popular Lookup Table. We abstained from evaluating layer-wise predictors (e.g. Wess et al. (2021)) since such data is not available, and meta-learning predictors (Lee et al., 2021) due to the vast possibilities to configure them. A separate and specialized comparison between classic and meta-learning predictors seems favorable to us.

**Simulation limitations**. In contrast to evaluating real predictors, the simulation allows us to quickly make statements for any test set sizes and predictor-inaccuracies. However, naturally, the results are only approximations. While they match actual values, they are generally slightly pessimistic. We also limit the simulation to HW-NAS-Bench since the changes to classification results are more accessible to interpretation than changes to loss values across different problem types. Finally, the current simulation approach does not consider specific architecture selection methods but an upper bound of always selecting optimally. This choice highlights that both the average accuracy and the hypervolume will be reduced for inaccurate predictors independently of the used optimization method. Nonetheless, a detailed study of how individual methods approach the upper bound is an interesting direction for future research.

**Broader Impact**. We hope that our study helps understanding and improving AutoML in general. Still, even with improved device-aware results, we do not believe AutoML to make ML engineering obsolete in the foreseeable future. For better or worse, AutoML is however capable of accelerating and magnifying the impact of many other machine learning fields.

**Transferability of the results**. Our evaluation includes five challenging and diverse datasets based on the micro-level search space of HW-NAS-Bench and five latency-based datasets of various macro-level search space architectures in TransNAS-Bench-101. Nonetheless, we find shared trends: All tested prediction models improve over Lookup Tables with little amounts of training data. Furthermore, most predictors benefit from more training data, even until the entire search space (aside from the test set) is known. We also find that network-based predictors are generally best but may be challenged by tree-based predictors if enough training data is available. Given only a few samples, Ridge Regression performs better than most other models.

**Recommendations**. While Lookup Tables are a cheap, simple, and popular model in gradient-based architecture selection, we find a significant variance in performance across tasks and devices (see Table 1). We recommend replacing such models with either MLPs or Ridge Regression, which are more stable, fully differentiable, and often take less than 100 training samples to achieve better results.

For most realistic scenarios where more than 100 training samples are available, MLP models are the most promising. They are among the top predictors on HW-NAS-Bench and demonstrate outstanding performance on the TransNAS-Bench-101 datasets. We found that specialized architecture encodings are primarily beneficial for little training data but suspect that they enjoy an additional advantage when network topologies are more complex and diverse (White et al., 2021).

While the query time for all predictors is less than 0.05s and thus negligible, there is a notable difference in training time (see the supplementary material). We recommend Ridge Regression for tiny amounts of training data and LGBoost otherwise if that is an essential factor.

If a NAS method selects architectures based on hardware metric predictions, we strongly suggest verifying the results by measuring the true metric value afterward. Doing so may eliminate

inferior candidates and improve the average result substantially. Finally, if the limiting factor to a particular NAS method is the slow live measurement of hardware metrics, using a much faster predictor may lead to an improvement, even if the prediction model is less accurate.

## 7 Conclusions

This work evaluated various hardware-metric prediction models on ten problems of different metrics, devices, and network architecture types. We then simulated the selection process for different test set sizes and predictor inaccuracies to improve our understanding of predictor-based architecture selection. We show quantitatively that even imperfect predictors may improve NAS if their low query time enables considering more candidate architectures. Finally, verifying the predictions for the few selected candidates can prevent using inferior architectures that are wrongly believed to be better, notably improving the average performance. The code and results are made available, thus acting both for recommendation and as a baseline for future works.

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
