## A  Best practices for NAS, Code and Data

To improve the reproducibility and facilitate fair experimental comparisons, we follow the best-practices checklist (Lindauer and Hutter, 2020):

- **Release Code for the Training Pipeline(s) you use**. Our experiments are based on White et al. (2021), who use NASLib (Ruchte et al., 2020) to compare 31 methods for accuracy prediction. Our NASLib fork, extending the framework for HW-NAS-Bench, TransNAS-Bench, some performance predictors and the hypervolume simulations, is provided in the supplementary materials. We intend to either make our fork available on GitHub or submit the changes upstream once this paper is accepted/published.

  It is noteworthy that we chose to explicitly normalize input data values (e.g. latency or energy consumption) for all predictors and datasets, which reduces the dependency of hyper-parameters (e.g. learning rate) on the dataset and allows us to analyze and compare the prediction errors across datasets more effectively.

- **Use the Same Evaluation Protocol for the Methods Being Compared**. Aside from the implementation of each predictor, all experiments use the same pipeline.

- **Validate The Results Several Times**. We ran each predictor 50 times, with seeds $\{0, ..., 49\}$. The reductions in hypervolume are simulated 1000 times using different a different subset of the data set, for each combination of {iteration, HW-NAS data set, noise on HW metric}.

- **Control Confounding Factors**. While all experiments used the same software libraries and hardware resources, they were run on different machines to speed up the evaluation. We found hardly any benefit in using a GPU even for the network-based predictors, which is why every method only used two CPU cores. Single experiments vary in time, usually ranging from few minutes to an hour. The OS is Ubuntu 18.04, notable software packages are PyTorch 1.9.0, numpy 1.19.5, scikit-learn 0.24.2, pybnn 0.0.5, ngboost 0.3.11, and xgboost 1.4.2

- **Report the Use of Hyperparameter Optimization**. See Appendix C.

  In addition to the code in the supplementary materials, we also provide the experimental results as csv files. Running the predictors and hypervolume simulations takes some time, but the easy access to the data of the finished experiments may prove useful for future research. Please see *readme.md* in the accompanying code zip file for instructions.

  We used the following data/assets/code:

- HW-NAS-Bench: `https://github.com/RICE-EIC/HW-NAS-Bench`, MIT License.

- TransNAS-Bench-101: `https://github.com/kmdanielduan/TransNASBench`, the repository and website lack an explicit license statement.

- NASLib: `https://github.com/automl/NASLib`, Apache License

## B  Encodings and Predictors

### B.1  Data encodings

Every architecture $a \in \mathcal{A}$ requires a unique representation, which depends on the used predictor. The common encoding types are:

   **Adjacency one-hot**: Each architecture $a$ is uniquely defined by the chosen candidate operation on every path. For example, each architecture in NAS-BENCH-201 consists of a repeated cell structure, which has five candidate operations on each of the six paths. Therefore there are

$5^6 = 15625$ unique architectures, which can each be referenced by a sequence of operation-indices such as [0 1 2 3 4 0]. Many predictors perform better if the sequence is presented as a one-hot encoding, which is in this case [10000 01000 00100 00010 00001 10000].

Similarly, the **path-encoding** (used by BANANAS) is a one-hot representation over the used candidate operation all possible paths. Since the connectivity within cells for HW-NAS-Bench and TransNAS-Bench-101 is fixed, it provides no more information than the adjacency one-hot encoding. If the connectivity can be adjusted more freely, as in the NAS-Bench-101 search space, the additional information may improve the fit.

The encodings for **BONAS**, **GCN**, and **NAO** each provide further information in addition to the Adjacency one-hot vector, most notably the adjacency-matrix. This $\{0, 1\}^{(N+2) \times (N+2)}$ matrix lists describes which of the $N$ architecture paths (rows) serves as inputs for each other path (column), and also includes input/output.

### B.2 Predictors

We briefly describe the 18 predictor methods in our experiments. We adopt their implementations from the NASLib library (see Appendix A), which we extend with Linear Regression, Ridge Regression, and Support Vector Machines from the scikit-learn package; and a simple Lookup Table implementation. Unless specified otherwise, the methods use the adjacency one-hot encoding.

- **BANANAS** An ensemble of three MLP models with five to 20 layers, each using the path-encoding (White et al., 2019).

- **Bayesian Linear Regression** A bayesian model that assumes (1) a linear dependency between inputs and outputs, and (2) that the samples are normally distributed (Bishop, 2007).

- **BOHAMIANN** A bayesian inference predictor using stochastic gradient Hamiltonian Monte Carlo (SGHMC) to sample from a bayesian neural network (Springenberg et al., 2016).

- **BONAS** Bayesian Optimization for NAS (Shi et al., 2020) uses a GCN predictor within an outer loop of bayesian optimization, as a meta-learning task. The GCN requires encoding the adjacency matrix of each architecture.

- **Gaussian Process** A simple model that assumes a joint Gaussian distribution underlying the training data (Rasmussen, 2003).

- **GCN** A Graph Convolutional Network that makes use of an adjacency-matrix encoding of each architecture (Wen et al., 2020).

- **Linear Regression** A simple model that assumes an independent value/cost for each operation/layer, which only need to be summed up. Unlike the Lookup Table model, it uses a least-square fit on the training data.

- **Lookup Table** The most simple and perhaps widely used model for differentiable architecture selection. It generally assumes a single baseline architecture (e.g. [001 001] in adjacency one-hot encoding), and a lookup matrix $\mathbb{R}^{(\text{num layers}) \times (\text{num candidates})}$ that contains the increases/reductions in the metric for each layer and candidate operation. The metric value of a new architecture can be predicted with a simple sum over the respective matrix entries and the baseline value. The model is obtained from measuring either each candidate operation in isolation, or by computing the differences between the baseline architecture and specific variations (e.g. [010 001] or [100 001], to measure the first candidates). This model always requires 1+(*num layers*) · (*num candidates*−1) neighbored architectures to fit. We detail the resulting correlation values for each used dataset in Appendix E.

- **LGBoost** Light Gradient Boosting Machine (LightGBM or LGBoost, Ke et al. (2017)) is a lightweight gradient-boosted decision tree model.

- **MLP** We use fully-connected Multi Layer Perceptrons in two size-categories.

- **NAO** (Luo et al., 2018) uses an encoder-decoder topology, which encodes/compresses an architecture to a continuous representation, and decodes it again. This representation is further used to make architecture predictions.

- **NGBoost** Natural Gradient Boosting (NGBoost, Duan et al. (2020)) is a gradient-boosted decision tree model that uses natural gradients to estimate uncertainty.

- **Ridge Regression** Ridge Regression (Saunders et al., 1998) extends the Linear Regression least-squares fit with a regularization term that serves as bias-variance tradeoff.

- **Random Forests** An ensemble of decision trees (Liaw et al., 2002).

- **Sparse Gaussian Process** an approximation of Gaussian Processes that summarizes training data (Candela and Rasmussen, 2005).

- **Support Vector Machine** A model that maps its inputs to a high-dimensional space, where training samples are used as support-vectors for decision-boundaries (Cortes and Vapnik, 1995).

- **XGBoost** eXtreme Gradient Boosting (XGBoost, Chen and Guestrin (2016)) is a gradient-boosted decision tree model.

## C  Hyperparameters

The default hyperparameters of the used predictors vary significantly in their levels of hyperparameter tuning, especially in the context of NAS. Additionally, some predictors may internally make use of cross-validation, while others do not. Following White et al. (2021), we attempt to level the playing field by running a cross-validation random-search over hyper-parameters each time a predictor is fit to data. Each search is limited to 5000 iterations and a total run time of 15 minutes and naturally excludes any test data.

We list our default and hyper-parameter sample ranges in Table 2. For comparability with White et al. (2021), we only change the values of newly introduced parameterized predictors: Ridge Regression, Support Vector Machines, and small MLPs.

Table 2: Hyper-parameter ranges and default values of the configurable predictors

| Model | Hyper-parameter | Range/Choice | Log-transform | Default |
|---|---|---|---|---|
| BANANAS | Num. Layers | [5, 25] | false | 20 |
| | Layer width | [5, 25] | false | 20 |
| | Learning rate | [0.0001, 0.1] | true | 0.001 |
| BONAS | Num. Layers | [16, 128] | true | 64 |
| | Batch size | [32, 256] | true | 128 |
| | Learning rate | [0.00001, 0.1] | true | 0.0001 |
| GCN | Num. Layers | [64, 200] | true | 144 |
| | Batch size | [5, 32] | true | 7 |
| | Learning rate | [0.00001, 0.1] | true | 0.0001 |
| | Weight decay | [0.00001, 0.1] | true | 0.0003 |
| LGBoost | Num. leaves | [10, 100] | false | 31 |
| | Learning rate | [0.001, 0.1] | true | 0.05 |
| | Feature fraction | [0.1, 1] | false | 0.9 |
| MLP (small) | Num. layers | [2, 5] | false | 3 |
| | Layer width | [16, 128] | true | 32 |
| | Learning rate | [0.0001, 0.1] | true | 0.001 |
| | Activation function | {relu, tanh, hardswish} | | relu |
| MLP (huge) | Num. layers | [5, 25] | false | 20 |
| | Layer width | [5, 25] | false | 20 |
| | Learning rate | [0.0001, 0.1] | true | 0.001 |
| NAO | Num. layers | [16, 128] | true | 64 |
| | Batch size | [32, 256] | true | 100 |
| | Learning rate | [0.00001, 0.1] | true | 0.001 |
| NGBoost | Num. estimators | [128, 512] | true | 64 |
| | Learning rate | [0.001, 0.1] | true | 0.081 |
| | Max depth | [1, 25] | false | 6 |
| | Max features | [0.1, 1] | false | 0.79 |
| Ridge Regression | Regularization $\alpha$ | [0.25, 2.5] | false | 1.0 |
| Random Forests | Num. estimators | [16, 128] | true | 116 |
| | Max features | [0.1, 0.9] | true | 0.17 |
| | Min samples (leaf) | [1, 20] | false | 2 |
| | Min samples (split) | [2, 20] | true | 2 |
| Support Vector Machine | Regularization $C$ | [0.5, 1.5] | false | 1.0 |
| | Kernel | {linear, poly, rbf, sigmoid} | | rbf |
| XGBoost | Max depth | [1, 15] | false | 6 |
| | Min child weight | [1, 10] | false | 1 |
| | Col sample (tree) | [0, 1] | false | 1 |
| | Learning rate | [0.001, 0.5] | true | 0.3 |
| | Col sample (level) | [0, 1] | false | 1 |

## D  NAS-Bench-201 / HW-NAS-Bench cell design

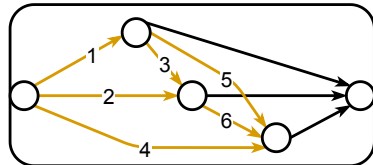

Figure 5: Basic NAS-Bench-201 / HW-NAS cell design. Each of the six orange paths is finalized with exactly one out of five candidate operations {Zero, Skip, Convolution 1×1, Convolution 3×3, Average Pooling 3×3}.

## E  Selection of datasets

Table 3: Kendall's Tau test correlation for Linear Regression, XGBoost, and Lookup Tables (LUT) on all HW-NAS-Bench datasets (rows), for different amounts of available training data (columns), tested on the remaining 625 samples. The Lookup Table model is tested on all 15625 architectures. We selected the five data sets at the top.

| | Linear Regression | | | | | | | | | | XGBoost | LUT |
|---|---|---|---|---|---|---|---|---|---|---|---|---|
| | 11 | 25 | 55 | 124 | 276 | 614 | 1366 | 3036 | 6748 | 15000 | 15000 | - |
| ImageNet16-120-raspi4_latency | 0.324 | 0.205 | 0.606 | 0.676 | 0.705 | 0.716 | 0.715 | 0.723 | 0.728 | 0.729 | 0.757 | 0.443 |
| cifar100-pixel3_latency | 0.392 | 0.292 | 0.732 | 0.780 | 0.797 | 0.803 | 0.806 | 0.809 | 0.812 | 0.812 | 0.877 | 0.484 |
| cifar10-edgegpu_latency | 0.370 | 0.258 | 0.724 | 0.790 | 0.806 | 0.819 | 0.820 | 0.822 | 0.830 | 0.829 | 0.926 | 0.175 |
| cifar100-edgegpu_energy | 0.376 | 0.275 | 0.732 | 0.793 | 0.812 | 0.821 | 0.821 | 0.823 | 0.831 | 0.831 | 0.920 | 0.221 |
| ImageNet16-120-eyeriss arith. int. | 0.369 | 0.293 | 0.748 | 0.805 | 0.817 | 0.827 | 0.825 | 0.832 | 0.843 | 0.846 | 0.970 | 0.861 |
| cifar10-pixel3_latency | 0.388 | 0.300 | 0.733 | 0.780 | 0.797 | 0.805 | 0.805 | 0.810 | 0.813 | 0.813 | 0.878 | 0.475 |
| cifar10-raspi4_latency | 0.393 | 0.315 | 0.740 | 0.787 | 0.799 | 0.805 | 0.807 | 0.810 | 0.813 | 0.813 | 0.890 | 0.462 |
| cifar100-raspi4_latency | 0.393 | 0.308 | 0.744 | 0.786 | 0.801 | 0.807 | 0.810 | 0.810 | 0.814 | 0.814 | 0.888 | 0.445 |
| ImageNet16-120-pixel3_latency | 0.398 | 0.312 | 0.739 | 0.786 | 0.799 | 0.807 | 0.809 | 0.812 | 0.815 | 0.816 | 0.884 | 0.509 |
| cifar100-edgegpu_latency | 0.375 | 0.268 | 0.728 | 0.793 | 0.810 | 0.821 | 0.820 | 0.822 | 0.831 | 0.831 | 0.924 | 0.191 |
| cifar10-edgegpu_energy | 0.375 | 0.284 | 0.728 | 0.792 | 0.810 | 0.821 | 0.823 | 0.824 | 0.831 | 0.831 | 0.922 | 0.183 |
| ImageNet16-120-edgegpu_energy | 0.377 | 0.281 | 0.733 | 0.797 | 0.814 | 0.825 | 0.825 | 0.826 | 0.834 | 0.833 | 0.926 | 0.280 |
| ImageNet16-120-edgegpu_latency | 0.379 | 0.264 | 0.737 | 0.799 | 0.817 | 0.826 | 0.826 | 0.828 | 0.836 | 0.835 | 0.938 | 0.277 |
| cifar10-eyeriss arith. int. | 0.384 | 0.296 | 0.757 | 0.811 | 0.826 | 0.835 | 0.832 | 0.843 | 0.854 | 0.854 | 0.969 | 0.826 |
| cifar100-eyeriss arith. int. | 0.384 | 0.297 | 0.757 | 0.811 | 0.826 | 0.835 | 0.833 | 0.844 | 0.855 | 0.856 | 0.971 | 0.830 |
| ImageNet16-120-fpga_latency | 0.443 | 0.494 | 0.904 | 0.936 | 0.947 | 0.951 | 0.948 | 0.951 | 0.952 | 0.952 | 0.983 | 0.965 |
| ImageNet16-120-fpga_energy | 0.443 | 0.494 | 0.905 | 0.935 | 0.947 | 0.951 | 0.948 | 0.951 | 0.952 | 0.952 | 0.983 | 0.965 |
| ImageNet16-120-eyeriss_latency | 0.457 | 0.937 | 0.953 | 0.954 | 0.954 | 0.954 | 0.953 | 0.953 | 0.954 | 0.954 | 0.952 | 0.989 |
| cifar10-eyeriss_latency | 0.461 | 0.943 | 0.959 | 0.959 | 0.960 | 0.960 | 0.959 | 0.960 | 0.960 | 0.960 | 0.958 | 0.995 |
| cifar100-eyeriss_latency | 0.462 | 0.946 | 0.963 | 0.963 | 0.963 | 0.963 | 0.963 | 0.963 | 0.964 | 0.963 | 0.962 | 0.998 |
| cifar10-eyeriss_energy | 0.456 | 0.967 | 0.985 | 0.985 | 0.985 | 0.985 | 0.985 | 0.985 | 0.985 | 0.985 | 0.975 | 0.996 |
| ImageNet16-120-eyeriss_energy | 0.458 | 0.967 | 0.985 | 0.985 | 0.986 | 0.985 | 0.986 | 0.985 | 0.985 | 0.986 | 0.972 | 0.998 |
| cifar100-eyeriss_energy | 0.457 | 0.967 | 0.985 | 0.985 | 0.985 | 0.986 | 0.985 | 0.986 | 0.986 | 0.986 | 0.976 | 0.998 |
| cifar10-fpga_energy | 0.458 | 0.973 | 0.987 | 0.987 | 0.987 | 0.987 | 0.987 | 0.987 | 0.987 | 0.987 | 0.986 | 0.999 |
| cifar100-fpga_energy | 0.458 | 0.973 | 0.987 | 0.987 | 0.987 | 0.987 | 0.987 | 0.987 | 0.987 | 0.987 | 0.986 | 0.999 |
| cifar100-fpga_latency | 0.457 | 0.973 | 0.987 | 0.987 | 0.987 | 0.987 | 0.987 | 0.987 | 0.987 | 0.987 | 0.986 | 0.999 |
| cifar10-fpga_latency | 0.457 | 0.973 | 0.987 | 0.987 | 0.987 | 0.987 | 0.987 | 0.987 | 0.987 | 0.987 | 0.986 | 0.999 |

Table 4: Kendall's Tau test correlation for Linear Regression and XGBoost on the five used TransNAS datasets (rows), for different amounts of available training data (columns), tested on the remaining 256 samples. The Lookup Table model (LUT) is tested on all 3256 architectures.

| | Linear Regression | | | | | | | | | | XGBoost | LUT |
|---|---|---|---|---|---|---|---|---|---|---|---|---|
| | 9 | 18 | 34 | 65 | 123 | 234 | 442 | 837 | 1585 | 2999 | 2999 | - |
| jigsaw | 0.201 | 0.227 | 0.410 | 0.535 | 0.586 | 0.605 | 0.616 | 0.624 | 0.631 | 0.632 | 0.661 | 0.201 |
| class_object | 0.268 | 0.262 | 0.518 | 0.646 | 0.711 | 0.741 | 0.759 | 0.771 | 0.780 | 0.780 | 0.828 | 0.701 |
| room_layout | 0.275 | 0.271 | 0.527 | 0.653 | 0.721 | 0.753 | 0.768 | 0.780 | 0.789 | 0.789 | 0.896 | 0.685 |
| class_scene | 0.275 | 0.268 | 0.527 | 0.653 | 0.721 | 0.755 | 0.768 | 0.782 | 0.789 | 0.790 | 0.907 | 0.710 |
| segmentsemantic | 0.282 | 0.259 | 0.545 | 0.684 | 0.746 | 0.780 | 0.798 | 0.809 | 0.816 | 0.818 | 0.871 | 0.726 |

**HW-NAS-Bench**:. To select five datasets that are (1) non-linear and (2) different from one another, we first fit Linear Regression to every available dataset, with the results listed in Table 3. The bottom 12 datasets can be accurately fit with only 25 training samples, so they are not very interesting as a challenge. On these datasets, the Lookup Table model achieves exceptional performance. Since the networks for CIFAR10, CIFAR100 and ImageNet16-120 only differ slightly, their measurements on the same device and metric (e.g. raspi4 latency) is very similar. To improve the generalizability of our results, we thus select datasets on different devices and metrics, which are listed at the top of Table 3. As displayed in Figure 6, their data distributions are generally different.

**TransNAS-Bench-101**:. As shown in Figure 7, the latency measurements of the architectures is generally very similarly distributed. We evaluate the possibly redundant datasets nonetheless, since latency predictions in macro-level search spaces are an important domain for NAS on image classification and object detection tasks. We select all data sets that provide the *test_loss* and *inference_time* attributes for all architectures, resulting in exactly the five datasets listed in Section 4 (the other two datasets contain more specific test losses).

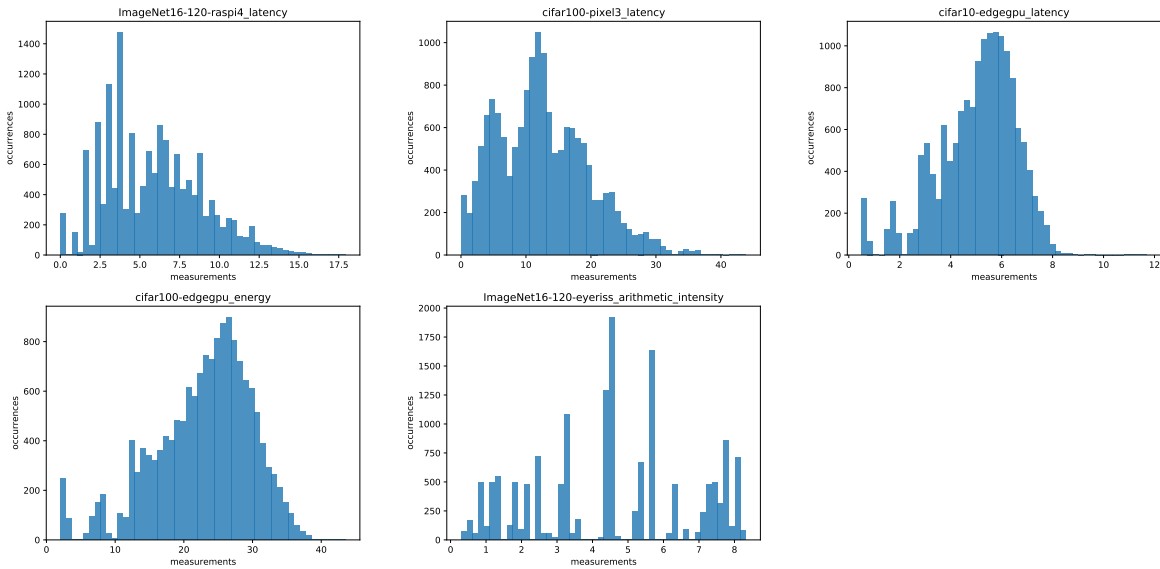

Figure 6: How the data of each selected HW-NAS-Bench dataset is distributed (not normalized).

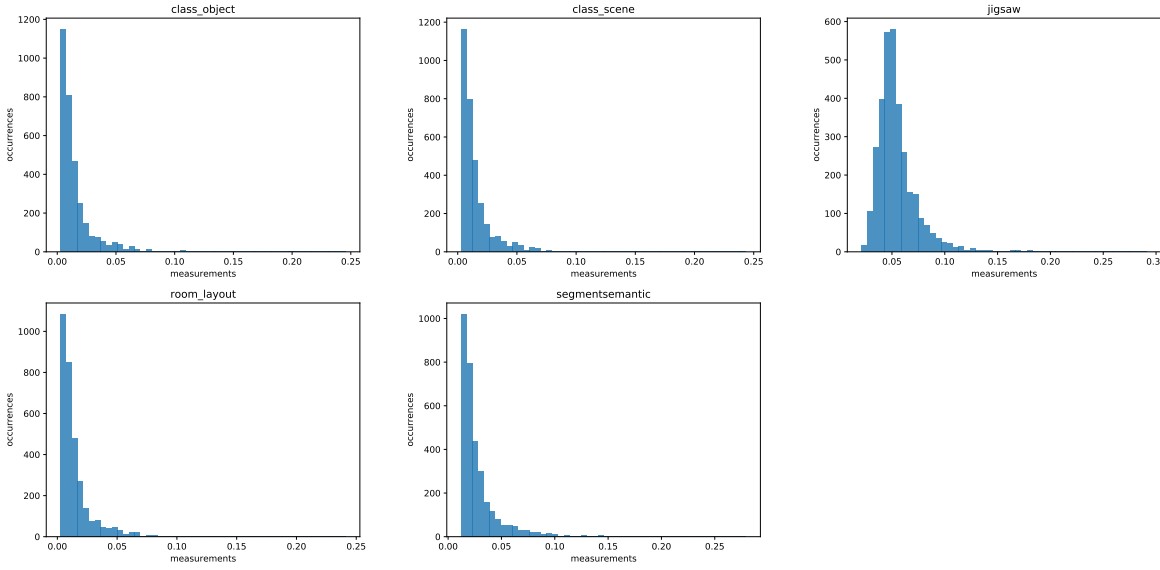

Figure 7: How the data of each selected TransNAS-Bench-101 dataset is distributed (not normalized). Since all architectures are measured for latency on the same hardware, the resulting datasets are much less diverse than the HW-NAS-Bench ones.

## F Predictor fit time

We visualize the average fit time of prediction models in Figure 8. Note that this may include a brief hyper-parameter optimization of up to 15minutes.

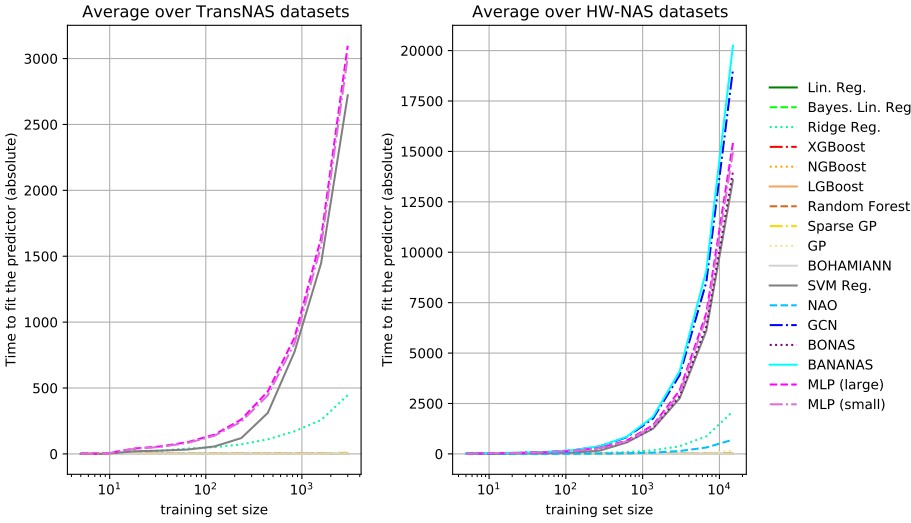

Figure 8: Fit time (in seconds) of predictors to data, depending on the training set size. By far the most expensive methods are network-based. However, a significant portion of this time is spent on the hyper-parameter optimization prior to the actual fitting.

# G Predictor deviations

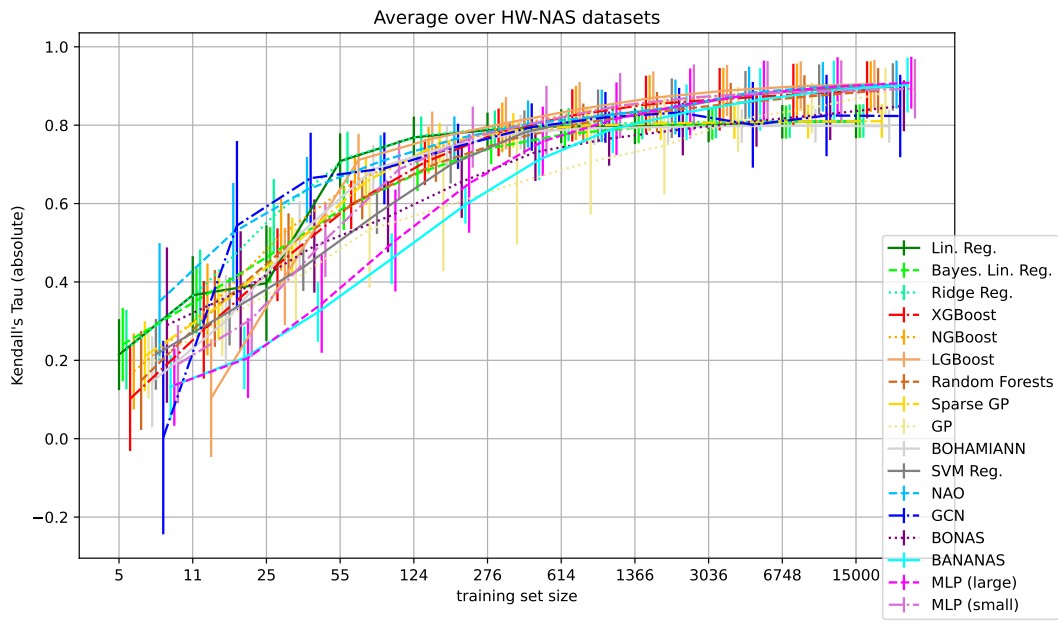

(a) Results on HW-NAS-Bench.

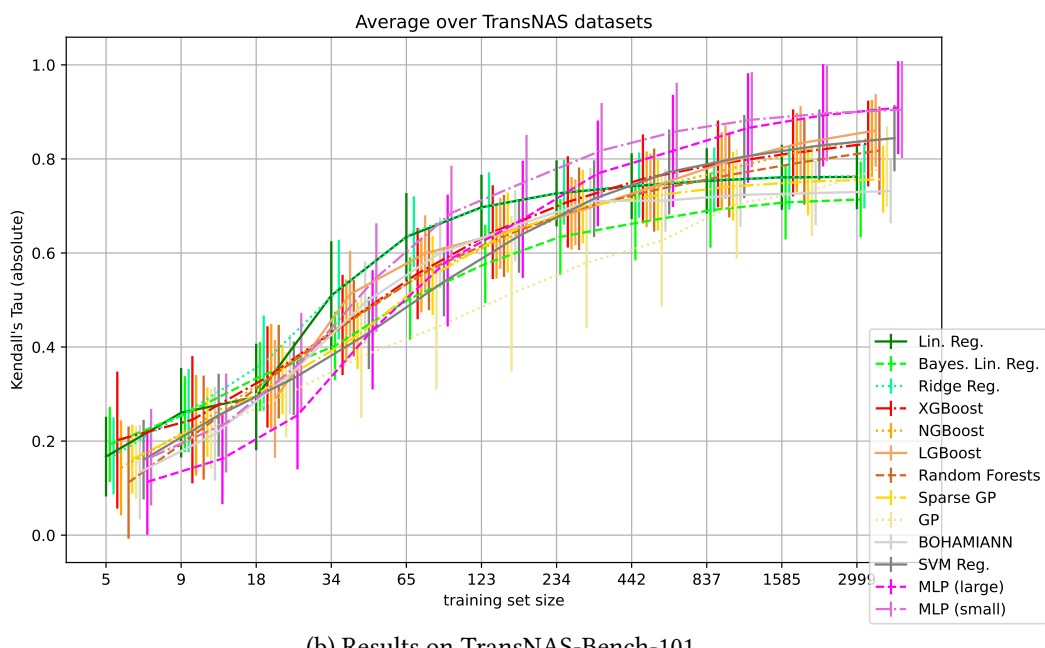

(b) Results on TransNAS-Bench-101.

Figure 9: As Figure 1, including the standard deviations of each set of predictors. Their slightly moved positions on the x axis are only for visual clarity. Note that for each training set size we sampled 50 different subsets of the full training data (using the random seeds) and have trained each type of predictor exactly once on each subset. The error bars likely reflect the effect of different subsets much more than that of potentially stochastic models/training.

## H  Approximating predictor mistakes

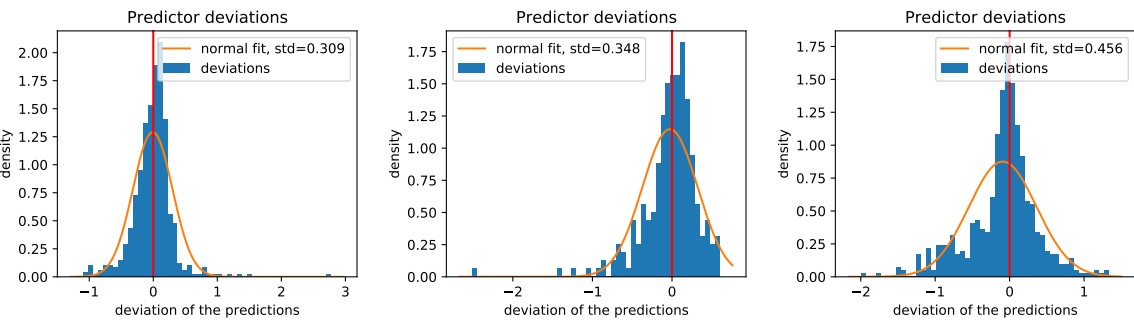

Figure 10: Further examples of predictor deviation distributions, as visualized in the center of Figure 2. **Left**: Linear Regression on CIFAR100, edgegpu, energy consumption. **Center**: Support Vector Machine on Jigsaw. **Right**: small MLP on ImageNet16-120, raspi4, latency.

Intuitively, the predictor deviation distributions (see Figures 2 and 10) generally resemble a normal distribution. However, most predictors:

(1)  Have a notable peak, sometimes off-center (e.g. at x=0.2)

(2)  Have less density than a normal distribution almost everywhere else

(3)  Have some outliers (e.g. at x>1.5) that are extremely unlikely for a normal distribution

Table 5: P-values of different distributions, trying to fit the distribution of all predictor mistakes according to a t-test. Larger values are better, but comparing many empirically sampled points with a true density function tends to push the p-values to 0.

|         | p-value |
|---------|---------|
| normal  | 0.028   |
| cauchy  | 0.030   |
| lognorm | 0.028   |
| t       | 0.028   |
| uniform | 0.037   |

We measured the p-value for different distributions on the first 100 test samples using a T-Test, every time we evaluated a predictor. The average statistics can be found in Table 5. Since a large number of empirical observations generally pushes the p-value to 0, this only serves to compare them to each other. We find that the outliers (3) appear often enough and are so unlikely to happen for a normal distribution, that even a uniform distribution has a higher statistical support. Consequentially, we approximate the common predictor deviations by sampling from a mixed distribution that adresses (1) to (3).

This mixed distribution consists of two Normal distributions ($N_1$, $N_2$) and one Uniform distribution ($U$), from which we sample with 72.5%, 26.5% and 1% respectively. For some constant $v$:

- We uniformly sample a shift $c$ from $[0, 2 \cdot v]$, that is used to push the centers of $N_1$ and $N_2$ to $x > 0$ and $x < 0$ respectively.

- We sample each value from $N_1(c, v)$, $N_2(-c, 3 \cdot v)$, and $U_1(-15 \cdot v, 15 \cdot v)$ randomly, with the weighting given above.

- We normalize (subtract mean, divide by standard deviation) our sampled distribution and then scale it to the desired standard deviation.

- The predictors produce non-smooth distributions. We simulate that by sampling 15 times fewer values as needed, and repeat them as often.

The code for the simulation is also provided (see Appendix A). As seen in Figure 11, the resulting simulated deviation distributions generally resemble a common predictor pattern. We do not account for differences in predictors, training set sizes or more, since that may become too specific and over-engineered.

Appendix J visualizes simulation sanity checks. We find that the simulation is slightly pessimistic and simplified, but resembles the results of actual predictors.

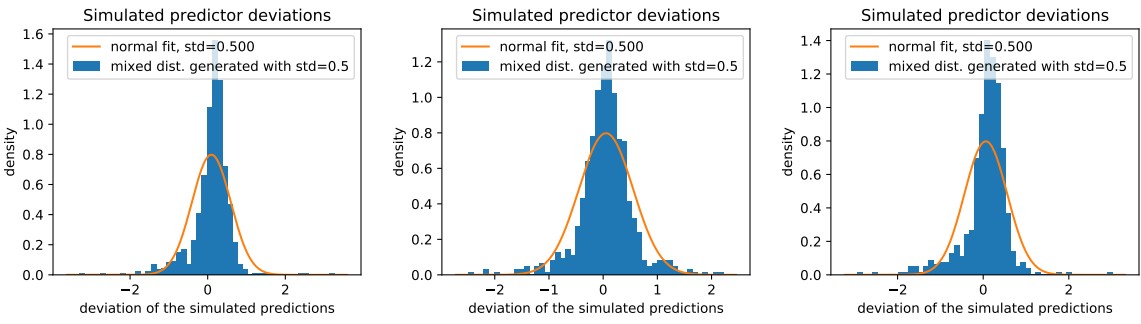

Figure 11: The sampled values of gaussian+uniform fit the measured predictor mistakes better than a single distribution, as they are roughly normally distributed, but include outliers.

# I Measuring simulated mistakes

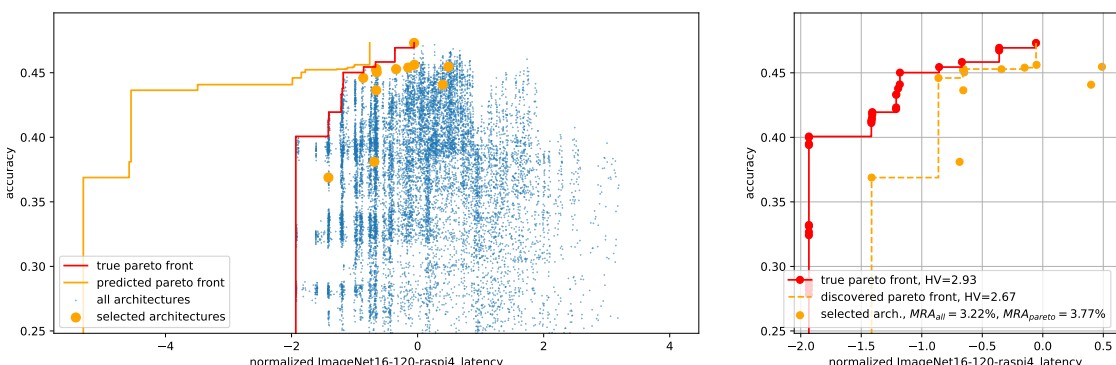

Figure 12: Similar to Figure 3. When the discovered Pareto set is considerably worse than the true Pareto set, it is possible for the Mean Reduction of Accuracy of the Pareto subset ($MRA_{pareto}$) to be *worse* than the average over all architectures ($MRA_{all}$). This naturally happens more frequently for worse predictors with a high sampling std. and low KT correlation. Consequentially, the difference between $MRA_{all}$ and $MRA_{pareto}$ is wider for better predictors (see Figure 4). Additionally, all of the selected non-Pareto-front members are clustered in a high-latency area and redundant with each other. This emphasizes the limitations of just considering drops in accuracy, as the hardware metric aspect is ignored. In this case, the predictor-guided selection failed to find a low-latency solution. Hypervolume solves these problems but is a less intuitive metric.

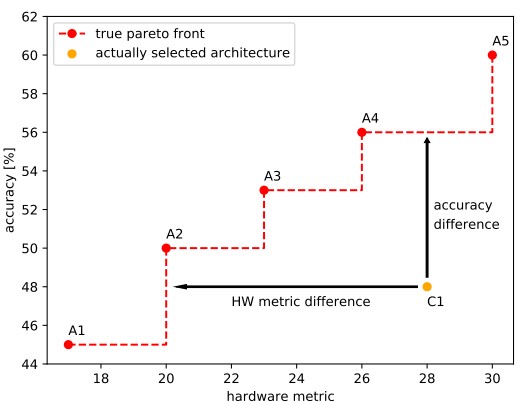 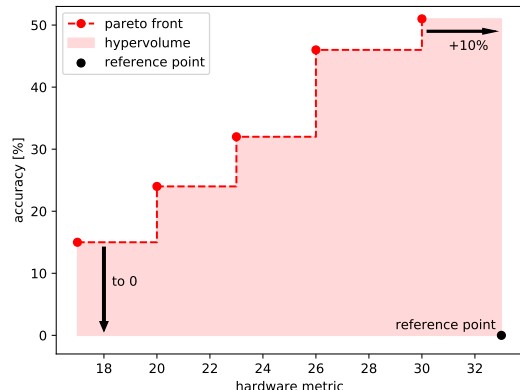

Figure 13: Examples to explain measurement methods.

**Left**: The distance of each selected candidate architecture C1 to the true Pareto front is measured, for accuracy and the hardware metric. C1 is dominated by A2, A3, and A4 of the true Pareto set. A2 has a slightly higher accuracy than C1 while being much better on the hardware metric, e.g. latency. A4 has a slightly better hardware metric value, but much higher accuracy. Given several candidate architectures, their differences are averaged.

**Right**: We compute the reference point for the hypervolume (for two objectives: area under a Pareto front) by multiplying the highest hardware metric value from the true Pareto front with 1.1, and accuracy 0. While we are consistent throughout all experiments, this choice is arbitrary, as there is no obviously correct choice for the reference point. If the hypervolume of a supposed Pareto front is computed, the reference point of the true Pareto front is reused. Thus, choosing inferior architectures will always reduce the hypervolume. We arbitrarily chose the multiplier of $m = 1.1$ as a middle ground between making the rightmost point of the Pareto front irrelevant ($m = 1.0$) and overemphasizing it ($m >> 1.0$).

## J  Simulation sanity check

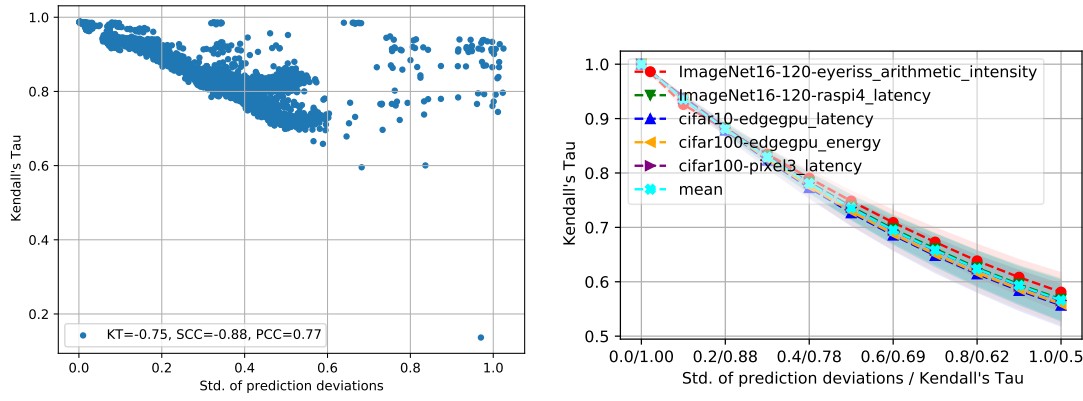

Figure 14: Standard deviation over the predictor deviations (x axis) and Kendall's Tau correlation (y axis), for the trained predictors on HW-NAS-Bench (left) and in simulation (right). The simulated predictor inaccuracies are slightly pessimistic (low KT), but still match the true values.

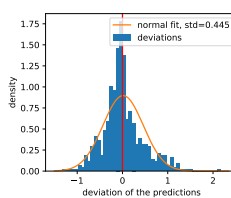

All candidates

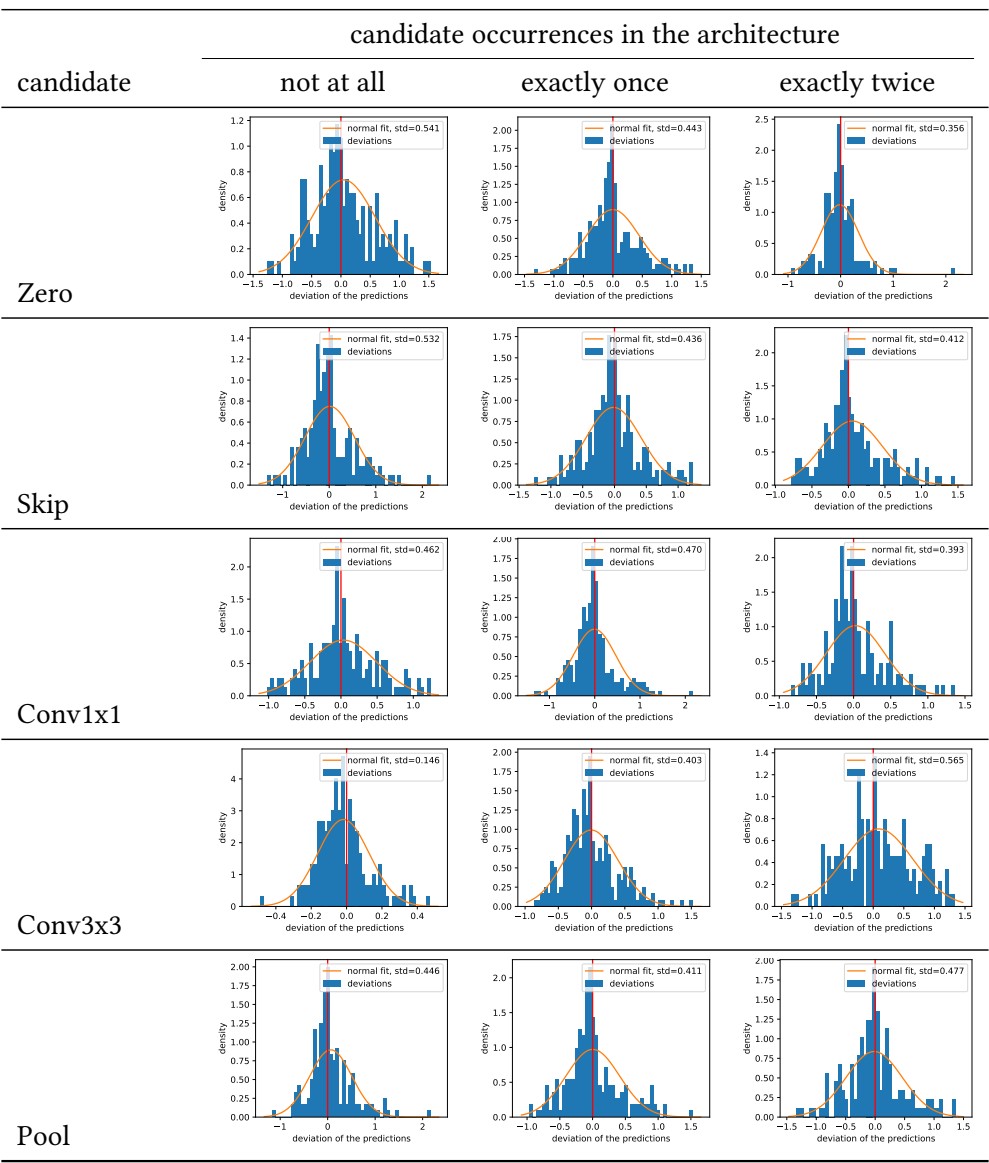

Table 6: How a trained XGB predictor deviates from the ground-truth values for different architecture subsets, akin to Figure 2. While they are not exactly the same, they still resemble the distribution over the entire test set (top plot, 625 samples). One noteworthy exception is when no Conv3x3 operations are used at all, in which case the standard deviation is considerably smaller.