# OpenReview forum: "What to expect of hardware metric predictors in NAS"
_automl.cc/AutoML/2022/Track/Main — AutoML-Conf 2022 (Main Track)_

### Official Review · Reviewer_bHA4 · 2022-04-04

**Potential Impact On The Field Of Automl Rating:** 4
**Technical Quality And Correctness Rating:** 3
**Clarity Rating:** 3

**Summary Of Contributions:**

This paper, inspired by the study by [White et al., 2021], empirically reviews the performance of different hardware metric predictors on HW-NAS-Bench and TransNAS-Bench-101. It further evaluates the effect of predictors on multiobjective/hardware-aware architecture search.

**Clarity:**

The paper is mostly well written and well explained. There are some questions needing clarifications:
-	In Figure 4, does each Kendall’s tau value on the x-axis correspond to a different predictor type? What’s the training data size for each predictor in Figure 4?
-	The final paragraph in section 5 is unclear to me (line 269 to 276). What does the equivalent set in line 270 refer to? Where can we draw the conclusion that a worse predictor leads to better results given the time for training a predictor is negligible compared to evaluating an architecture?

**Overall Review:**

Pros:
-	the extensive empirical study on different hard-ware metric predictors are useful for future hardware-aware NAS research
-	the empirical analyses are thorough and well structured. I appreciate the authors’ efforts to evaluate the effect of predictors on multi-objective search on top of comparing their fitting results, which gives better idea on how well predictors work in practice.
-	I quite like the discussion section in the paper which summarises the key insights and provide useful recommendations for the readers.


Cons:
-	The novelty of the paper is quite limited because only existing predictors are evaluated and the empirical analyses mainly follow the structure in [White et al., 2021]. It would make the paper more interesting if the authors can leverage the insights learnt and propose a new type of hybrid predictor as in [White et al., 2021].
-	The conclusion from the empirical analyses is kind of expected without much surprising insights.


**Potential Impact On The Field Of Automl:**

This paper will have good impact on the AutoML community because hardware-aware NAS is a very promising direction with large industrial interests. The insights and recommendations on which predictor to use for hardware metric estimation under different budget and circumstances will be useful for future hardware-aware NAS research.

**Reproducibility:**

The authors include good reproducibility details in the appendix and we should be able to reproduce the results.

**Review Confidence:**

5: You are absolutely certain about your assessment. You are very familiar with the related work and checked all the details carefully.

**Review Rating:**

5: Accept, good paper

**Review Summary:**

While the novelty of the paper is limited, such extensive empirical study on hard-ware metric predictors is still useful and necessary for NAS community. The authors do a good job in conducting this empirical study and in delivering/deriving the useful insights/recommendations. Thus, I would recommend to accept this paper.

**Technical Quality And Correctness:**

This paper mainly conducts empirical analyses and follows the thorough practices in [White et al., 2021]. Technical parts appear correct to me. However, I do have some questions that I would love to hear the authors’ responses:
-	As acknowledged by the authors, measuring hardware metrics is rather cheap as it often doesn’t necessitate any network training. Why do we need predictor for such metrics in the first place? If the potential advantage is to avoid measuring on the hardware during the search, don’t we still need to evaluate the architecture validation accuracy on the hardware device during the search. Could you clarify my misunderstanding here?
-	On fitting results and comparison, I’m quite curious why SVM scales well with more training samples but GP doesn’t given both are using kernels? and why SVM outperforms GP? Any intuition on this would be appreciated.
-	Why do we need to simulate the predictor deviations? Cannot we just modify the predictors into their Bayesian counterparts to get the predictive uncertainty ? In fact some predictors like BLR , GP and BOHAMIANN already outputs the predictive uncertainty.
-	Does all the predictors only take in architecture as the input? Would the predictor input also include hardware meta-features which will influence some hardware metrics?

---

### Official Review · Reviewer_xjLx · 2022-04-04

**Potential Impact On The Field Of Automl Rating:** 3
**Technical Quality And Correctness Rating:** 3
**Clarity:** The paper is very well written.
**Clarity Rating:** 4

**Summary Of Contributions:**

This paper presents a wide experimental evaluation of performance predictions in the context of NAS (but could be applicable more generally). The experiments show that MLP-based performance models are generally the most accurate, and on the opposite common-used lookup tables are the most inaccurate if compared to models trained on a reasonable amount of data. The authors then measure the discrepancy between model performance predictions and real performances. They show that for good predictors a subset of the best candidates of the pareto frontier identified using the performance predictions corresponds closely to the best candidates of the real pareto frontier. They measured this discrepancy with an hypervolume, the space between the estimated pareto frontier and the true pareto frontier.

**Overall Review:**

+ Is it a relevant topic for the conference
+ The message is clear and well supported

- Computational cost of performance estimation is minor compared to accuracy estimation and therefore the improvements possible are limited.

**Potential Impact On The Field Of Automl:**

Estimation is particularly expensive for AutoML routines that sit of top of already expensive training loops. For this reason model performance and accuracy estimation is becoming prevalent in AutoML which reduces substantially the benchmarking costs but increases the risk of misleading results. This work can be impactful in that it covers the reliability of model performance estimation and provide simple ways to improve upon them. As the authors mention however, performance estimation is nowhere as expensive as model training and thus model-based performance estimations is not as impactful as model-based accuracy estimation would be. I believe nevertheless that this is an important topic.

**Reproducibility:**

The reproducibility list is filled and the code shared seams to allow to reproduce the results although I did not verify myself. I would like to make a comment on the following question/answer:

(k) Did you report error bars (e.g., with respect to the random seed after running experiments multiple times)? [No] We lack the required space to do so for most plots. However, since all results and the plotting code are made available, we do not think this to be a problem.

It should not be the reader's responsibility to assess whether a result is statistically significant or not by running the experiments multiple times. There should be error bars.

**Review Confidence:**

3: You are fairly confident in your assessment. It is possible that you did not understand some parts of the submission or that you are unfamiliar with some pieces of related work.

**Review Rating:**

5: Accept, good paper

**Review Summary:**

I believe the message of this paper is relatively important. The experiments are broad supports well the message as well. The impact of this work may be limited however because performance estimation is quite inexpensive compared to accuracy estimation.

**Technical Quality And Correctness:**

The paper presents various model performance methods but does not detail the technical aspects, which I believe is fine.

The experiments are well designed and supports well the claims of the papers. There is however 2 points that I think could be improved.

The first one is the look-up table evaluation. Lookup tables are like histograms and therefore can be seen as non-parametric ML models too. So, considering the training budget for the lookup-table/histogram, how does it compare with the parametric models? I would assume the results to still favor the MLPs, but anyhow this would be a more truthful/fair comparison.

The second point is about the interpretation of the results, especially in figure 1. Kendall's rank correlation coefficient can have high variability on small number of samples and thus should be reported with error bars.

---

### Official Review · Reviewer_TJQ2 · 2022-04-05

**Potential Impact On The Field Of Automl Rating:** 3
**Technical Quality And Correctness Rating:** 4
**Clarity:** I found the paper easy to follow and …
**Clarity Rating:** 4

**Summary Of Contributions:**

This paper evaluates 18 different predictors for hardware metrics on 2 NAS benchmarks, NAS-Bench-201 and TransNAS-Bench-101. These predictors are compared based on the rank correlation and the authors investigate how the simulated Pareto-front based on these hardware metric predictors compares to the true Pareto-front. Their findings show that model-based predictors outperform lookup tables with few data points and that simple MLPs perform the best on average across all their evaluated settings.



**Overall Review:**

Positive:
- Useful insights on hardware metric predictors for NAS, which will be useful for a lot of researchers in the AutoML community.
- 18 different performance predictors are utilized for the analysis and are evaluated across 2 NAS tabular benchmarks.
- All the code is available on the same framework.
- The paper is well-structured and mostly easy to follow

Negative and other comments:
- Without belittling the importance of the large-scale evaluation conducted in this paper, the novelty aspect is lacking. To circumvent this my suggestion would be to go along the lines of for instance combining predictors as done in White et al. (2021) with the OMNI predictor.
- I would also be curious to see how some multi-objective or hardware-aware NAS algorithms perform in the 2 settings: (1) when using the predicted hardware metric and (2) using the true hardware metric.

**Potential Impact On The Field Of Automl:**

I think this paper provides insights on an important problem which will be relevant in particular for the hardware-aware NAS community. As far as I know, it is the first large-scale study of hardware metrics predictors on an unified codebase and evaluated on several benchmarks. Furthermore, since the implementation is in NASLib, it can go very well along with the other tools implemented in this framework and motivate the implementation of multi-objective NAS algorithms in NASLib.

**Reproducibility:**

All the benchmarks used in the paper are public and the paper's code is implemented in NASLib, which is an open-source tool for NAS research. The authors also provide a .zip file with all the necessary code to reproduce their experiments.

**Review Confidence:**

5: You are absolutely certain about your assessment. You are very familiar with the related work and checked all the details carefully.

**Review Rating:**

5: Accept, good paper

**Review Summary:**

In general, this paper provides some useful insights on hardware metric predictors in NAS and can be of practical relevance to the community since everything is implemented in NASLib and it can be used for benchmarking purposes in multi-objective NAS. I would therefore recommend acceptance.

**Technical Quality And Correctness:**

The analysis conducted in this paper is meaningful and insightful. I do not see any problems with the experiments and the conclusions drawn from their results. Most of the code is based on White et al. (2021), which has already implemented the performance predictors and shown in their Table 7 (Appendix) that most of the predictors closely yield similar results as in the corresponding original papers.

---

### Official Review · Reviewer_2Hpw · 2022-04-11

**Potential Impact On The Field Of Automl Rating:** 3
**Technical Quality And Correctness Rating:** 3
**Clarity Rating:** 4

**Summary Of Contributions:**

The paper provides a empirical analysis of the efficacy of several hardware predictors and their downstream impact on the end to end NAS algorithm performance on the HW-NAS-Bench and Trans-NAS-Bench-101 benchmarks.

The authors evaluate 18 prediction algorithms on several hardware and task scenarios from these benchmarks and provide some insights into the relative advantages and disadvantages of these predictors in different scenarios. The authors then evaluate the impact of predictor sub-optimality on the end output - comparing the discovered Pareto-front with the optimal Pareto-front. Using these experiments, the authors provide insights and recommendations on how to choose the best predictor for a given task.

**Clarity:**

The paper is overall well written and easy to follow. The figures (especially Fig.1, Fig.3, Fig.4) provide intuitive visualizations to back up the authors main points.

**Overall Review:**

Strengths :

1) With the increasing growth and focus on Efficient ML and Hardware-Aware NAS, understanding HW performance prediction is a fairly important problem. While there have been plenty of papers that study performance prediction for accuracy, this work is (to the best of my knowledge) the first work to study performance prediction specifically in the HW context.
2) The paper is well written and provides simple and actionable insights for future work backed by data and easy to understand visualizations.
3) The authors provide the code and a detailed appendix, making reproducibility and future work on this topic accessible.

Weaknesses :

1) Novelty : While this work provides some insights, the novelty is limited, considering that this work is mostly an adaptation of [1] to the hardware prediction domain. Some of insights presented as well (improved performance with more training data, efficacy of MLPs and better Pareto fronts with more samples evaluated) are not particularly new.

2) Transferability : (elaborated in the questions above).


**Potential Impact On The Field Of Automl:**

The paper provides a few insights on hardware predictor choices in future work in HW-Aware NAS. These are:
1) Neural Networks provide a strong and simple baseline for HW performance prediction, achieving high Kendall's Tau ranking correlation (KT correlation) even when trained on very few samples. While this is not surprising/novel and fairly intuitive, this study provides a data-driven point - considering the evaluation across 18 methods.

2) Lookup Tables, despite the wide usage in previous works, are sub-optimal in terms of overall KT correlation to most predictors trained even on 50-100 data points. This was a somewhat surprising result, and definitely is something to keep in mind for future research and applications. However, I have a question regarding this (mentioned below)

3) This work introduces a novel approach to evaluating the downstream impact of performance predictors on the final results of NAS. By creating a proxy distribution of a predictor error/deviation and simulating search along with a predictor, the authors quantify the overall drop/sub-optimality resulting from an inaccurate predictor.

I feel this is the most interesting contribution from the paper - it gives a picture of how close one can get to the optimal result given an evaluation budget and the best method to choose in this context. While this evaluation is empirical and based on the distribution estimate, (and not the actual predictors), it presents a first step in this direction


*Questions:*
I have a few questions about the transferablity/generalization of these insights/recommendations, about aspects that I feel significantly restrict the potential impact of this work :

a) Why is a particular HW space hard/easy? It is mentioned that FPGA/Eyeriss are suitable for Lookup Tables (and it seems pixel3 is not), but with no explanation (beyond KT ranking). While we can evaluate this in the context of pre-computed Hardware NN benchmarks, the challenge is in extending this insight to a new problem/hardware space.  While (2) is an important observation, this work does not make it exactly clear when (2) applied or not. Without this clarity, it's unclear of how to use this insight while dealing with a new problem.


b) How do predictors fare on other neural network spaces? Both HW-NAS-Bench, Trans-NAS-Bench-101 have a similar cell-based NN space and limited variability? It is not clear whether the same results will generalize to other non-cell based space (for example, the spaces in [1], [2], [3])

c) This work studies predictors mostly in an NAS-algorithm independent setting. Different algorithms use predictors differently - for example, a DNAS based algorithm might rely on gradients from the predictor (in which case high KT correlation might not be a good evaluation metric). Another example is : a predictor being used to guide the search process - in selecting which sub-network to train.
It's unclear how the insights will transfer in these contexts.

[1]ProxylessNAS: Direct Neural Architecture Search on Target Task and Hardware (https://arxiv.org/abs/1812.00332)

[2] FBNet: Hardware-Aware Efficient ConvNet Design via Differentiable Neural Architecture Search (https://arxiv.org/abs/1812.03443)

[3] Once-for-All: Train One Network and Specialize it for Efficient Deployment (https://arxiv.org/abs/1908.09791)

**Reproducibility:**

The results in the paper are easy to reproduce - the authors have shared the code used in the experiments, the raw results, and have provided comprehensive details of the experimentation methodology and hyper-parameters in the appendix. This work is based on publicly available libraries (NASLib) and benchmarks (HW-NAS-Bench, Trans-NAS-Bench-101).

**Review Confidence:**

4: You are confident in your assessment, but not absolutely certain. It is unlikely, but not impossible, that you did not understand some parts of the submission or that you are unfamiliar with some pieces of related work.

**Review Rating:**

4: Marginally above the acceptance threshold (use sparsely)

**Review Summary:**

This paper provides some interesting insights and analysis about a problem (HW Performance prediction) that is of importance to the AutoML community. However, the current insights lack depth and there are a few concerns regarding the transferablity/generalization of these recommendations. I currently give this paper a weak accept, but I'm willing to change my evaluation if the mentioned concerns are addressed.

**Technical Quality And Correctness:**

Overall, the approach and experimentation are technically correct and sound. The authors evaluate a large set of performance predictors on a variety of hardware and task settings. While this way of evaluation is not technically novel in itself (since it is essentially an adaptation of [1] to the problem of HW performance prediction), this is the first work to explore this direction. This work arrives at fairly logical conclusions and recommendations based on the experiments. However, I have a concerns about whether the recommendations can transfer beyond the given benchmarks and tasks (elaborated in the questions above)

[1] How Powerful are Performance Predictors in Neural Architecture Search?  (https://arxiv.org/abs/2104.01177)

---

### Official Review · Reviewer_NPBD · 2022-04-12

**Potential Impact On The Field Of Automl:** n/a
**Potential Impact On The Field Of Automl Rating:** 1
**Technical Quality And Correctness:** n/a
**Technical Quality And Correctness Rating:** 3
**Clarity:** n/a
**Clarity Rating:** 3

**Summary Of Contributions:**

n/a

**Overall Review:**

n/a

**Reproducibility:**

The authors provide a zip file containing their experiment code integrated into the naslib package.
The code further depends on HW-NAS-Bench and TransNAS-Bench-101 which can be found online in separate repositories.
For replicating the environment of the work, I created a conda environment with major dependencies, installed the package via ``pip install -e .`` through setuptools as mentioned and step-by-step added further dependencies until the mentioned *runner.py* executed without errors.
Following aspects would be good to improve replicatability:

- explain custom paths such as "/data/datasets/", which can be currently only found in python scripts under _hw/, in a central place such as the readme
- consider to work with conda, poetry or pip in an own repository such that dependencies like naslib and hw-nas-bench can be installed through one command and document it - it took quite some time to figure out all dependencies like *torch_sparse*
- provide some more examples of where the configuration files rely in and how to compose a script invocation such as ``python3 naslib/benchmarks/predictors/runner.py``
The replicatability can be considered as somewhat difficult and time-expensive, but possible.

 The reproducibility check list is filled properly.

**Review Confidence:**

4: You are confident in your assessment, but not absolutely certain. It is unlikely, but not impossible, that you did not understand some parts of the submission or that you are unfamiliar with some pieces of related work.

**Review Rating:**

5: Accept, good paper

**Review Summary:**

n/a

---

### Meta-Review · Area_Chair_QX8k · 2022-05-08

**Recommendation:** Accept
**Confidence:** 5

**Metareview:**

The authors conduct a comparison of several hardware predictors inspired by the work of White et al. 2021. All reviewers consider this study useful but also identified some areas that require deeper investigation. It remains unclear whether the results transfer to different search spaces or NAS algorithms that differentiate through the predictor. There is a clear agreement to accept this work at the conference.

---

### Decision · Program_Chairs · 2022-05-13

Accept